# Understanding the role of regulatory flexibility and context sensitivity in preventing burnout in a palliative home care team

**Vittorio Lenzo**[1]*, **Valentina Bordino**[2], **George A. Bonanno**[3], **Maria C. Quattropani**[1]

**1** Department of Clinical and Experimental Medicine, University of Messina, Messina, Italy, **2** Palliative home care services, Sisifo—Consortium of Social Cooperatives, Catania, Italy, **3** Teachers College, Columbia University, New York, NY, United States of America

* vittorio.lenzo@unime.it

**Data Availability Statement:** Data set file is attached to this revised submission.

## Abstract

Although burnout syndrome has been investigated in depth, studies specifically focused on palliative home care are still limited. Moreover, there is still a lack of evidence regarding the interplay between emotional flexibility and sensitivity to context in preventing burnout in home care settings. For these reasons, the aims of this study were to examine burnout symptoms among practitioners specializing in palliative home care and to investigate the role of regulatory flexibility and sensitivity to context in understanding burnout. An exploratory cross-sectional design was adopted. A convenience sample (n = 65) of Italian specialist palliative care practitioners participated in this study. Participants were recruited between February and April 2019 from two palliative home care services that predominantly cared for end-of-life cancer patients. The Italian version of the Maslach Burnout Inventory (MBI), the Flexible Regulation of Emotional Expression (FREE) scale (a measure of emotional flexibility), and the Context Sensitivity Index (CSI) (a measure of sensitivity to context) were administered. Analyses of variance were conducted using the three MBI factors as dependent variables and profession as an independent variable. Subsequently, three identical analyses of covariance were conducted with age, work experience, flexibility and sensitivity to context as covariates. The results showed a low burnout risk for all three of the MBI factors, and there were no gender differences. An ANOVA revealed a significant effect of profession type and age on the emotional exhaustion factor of the MBI, and an ANCOVA indicated that these effects persisted after covariates were accounted for. The results also showed a significant effect of the FREE score on emotional exhaustion. These findings can help explain the differential contributions of profession type and age to the burnout symptoms investigated. In addition, the emotional flexibility component, as an aspect of resilience, represents a significant and specific factor of emotional exhaustion. Interventions to prevent burnout must consider these relationships.

**Funding:** This research did not receive any specific grant from funding agencies in the public, commercial, or not-for-profit sectors.

**Competing interests:** The authors declare that no competing interests exist.

## Introduction

Burnout syndrome is defined as a state of emotional exhaustion, depersonalization, and reduced personal accomplishment, and this syndrome can occur among individuals in many human services professions [1]. Palliative care workers continuously care for patients and support their suffering relatives with the aim of improving their quality of life. Many studies have shown that burnout can negatively impact the quality of patient care and health-care systems [2]. To date, research on the prevalence of burnout among palliative care practitioners has produced contradictory results. Past studies have found a prevalence ranging from 17 percent to 62 percent, with most burnout stemming from emotional exhaustion [3–5]. Various risk factors exist for the development of burnout among palliative care practitioners, and several studies have documented considerable variability in sources of burnout across clinician types and practice settings [4, 6–8]. Previous studies have shown that working on a palliative care mobile team can lead to a higher risk of developing burnout and psychological morbidity (e.g., depression) than working in an actual palliative care unit [3, 9].

Synthesizing the aforementioned research findings is very difficult due to the range of burnout risk factors that have been examined. Investigating the differential role that these factors play in the palliative home care setting is also challenging. Practitioner- and setting-related variables have also impacted studies on the efficacy of interventions to prevent and reduce burnout symptoms [10–13]. Most of these studies have utilized samples of physicians and paid less attention to other health-related professions such as nurses, psychologists, social workers, or physiotherapists. However, the field of palliative care employs a multidimensional team approach to address the needs of patients and their families [14], and palliative care practitioners endure traumatic experiences that disrupt their sense of continuity of their own lives [15]. In addition, the absence of an accurate statistical model explaining burnout syndrome makes it problematic to ascribe the findings to specific factors. In light of this perspective, a recent study involving a sample of 185 palliative care nurses was the first to provide a model with good fit [16]. The authors demonstrated that psychological, work-related and demographic factors were associated with burnout.

In recent years, a growing number of studies have examined the role that resilience plays in the identification and reduction of burnout symptoms among palliative care practitioners [17–23]. Resilience is a multifaceted construct with different definitions in the context of individuals or organizations [24]. In the context of palliative care, however, the absence of a clear and consistent definition of resilience across studies has resulted in significant variability among assessment instruments. In this light, Bonanno and colleagues introduced the concept of expressive flexibility for understanding an individual's ability to enhance or suppress emotion across different contexts [25]. Bonanno explained resilience very simply as a stable trajectory of healthy functioning following highly adverse and stressful events. From the perspective of this framework, resilience reflects the ability to maintain a stable equilibrium after exposure to stressful and traumatic situations. Past theories and studies have mistakenly assumed that coping and emotion regulation strategies are always beneficial or maladaptive [26]. Therefore, adaptation depends on one's ability to flexibly enhance or suppress emotional expression in accordance with contextual demands [27]. The extent to which people possess these characteristics could explain how people respond to stressful events. Findings from several studies have indicated that regulatory flexibility represents a central component of mental health in healthy subjects and people after traumatic or adverse events [28–33]. Currently, however, little is known about the roles of expressive flexibility and sensitivity to context in the development of burnout symptoms among home palliative care practitioners. It is logical to assume that these components can play a role in reducing one's risk for experiencing burnout. Due to the

contradictory results on the prevalence of burnout among palliative care practitioners, it is useful to examine the differential risks for burnout among different practitioner types. As previously stated, most studies to date have focused on physicians; however, few researchers have specifically examined burnout in the palliative home care setting, where the burnout risk may be higher.

The first aim of this study was to compare burnout by gender in a sample of practitioners specializing in palliative home care. We hypothesized that there would be no significant differences between males and females. The second aim of this study was to examine the risk of burnout in a sample of practitioners specializing in palliative home care. We hypothesized a medium-to-high risk percentage for the three factors of burnout. The third aim of this study was to explore the relation between the type of professional practitioner and burnout risk. We hypothesized that we would find differences between the palliative home care specialists based on their different contributions to the treatment team. Finally, the fourth aim of this study was to examine the effect of expressive flexibility and sensitivity to context on the three factors of burnout. We hypothesized that these individual variables would significantly influence burnout risk.

## Materials and methods

### Participants

The study was conducted with a sample of 67 Italian practitioners specializing in palliative care and working in two palliative home care services in Sicily, Italy. Both palliative care services operate in the cities of Messina and Agrigento and are included in the Local Health Unit of the Italian National Health Service (INHS). All the participants worked in teams of two, and the palliative home care services were provided predominantly to end-of-life cancer patients. A plenary meeting was conducted to present the aims of the study and administer the self-report instruments. It was explained to participants that study enrolment was voluntary and that participation in the study would not influence the nature or conduct of their work. Before the beginning of the study, all participants gave written consent. Sixty-seven participants took part in the research, but 2 did not return the completed form at the end. However, the final sample represents 70 percent of the combined number of palliative care workers from the two services. The inclusion criteria were current employment in palliative home care services with a full-time contract and no prior affiliation with any aspect of the present study. The exclusion criterion was a positive anamnesis for any psychiatric disorder included in the Diagnostic and Statistical Manual of Mental Disorders (DSM-5) [34].

A post hoc power analysis, conducted using G*Power (version 3.1.9.4) [35], ensured that the statistical power was adequate to provide statistically significant results. Hence, statistical power was computed as a function of the significance level $\alpha$ and population effect size used in this study [36, 37]. For these reasons, we selected the F test and ANCOVA with fixed effects, main effects and interactions as settings in G*Power. Consequently, a statistical power of .88 (with a critical $F$ of 4.01) was obtained by entering a significant finding (at the .05 level), a large effect size (Cohen's $d$ = 0.40), a total sample size of 65, 6 groups, and 4 covariates.

### Procedure

The study was conducted in accordance with the 1964 Declaration of Helsinki and its later amendments. A self-report questionnaire was administered to collect data on age, gender, level of education, work experience, and professional role. The study was approved by the Research Ethics Committee for Psychological Research of the University of Messina (n. 93113).

## Measures

The following three self-report measures were administered.

1. The *Maslach Burnout Inventory* (MBI) is a 22-item self-report questionnaire assessing the symptoms of burnout [38, 39]. The items are rated on a 7-point Likert scale ranging from 0 (never) to 6 (every day). The questionnaire includes the following three subscales (the Cronbach's alpha scores derived from this sample are reported in parentheses): Emotional Exhaustion (EE; α = .85), which measures reduced energy and emotional and cognitive distancing from the job; Depersonalization (D; α = .73), which measures lack of engagement, cynicism, and distancing from patients; and Personal Achievement (PA; α = 77), which measures the ability to work well with others, the ability to effectively deal with problems at work, and the perception of having a positive influence on others. In the present study, a validated Italian version of the MBI was used [40]. Scores for the EE subscale can be categorized as follows: scores ≥ 23 indicate a high risk, while scores ranging from 14 to 22 indicate a medium risk. Regarding the DP subscale, scores ≥ 6 indicate a high risk, while scores ranging from 3 to 5 indicate a medium risk. Regarding the PA subscale, scores ≤ 31 indicate a high risk, scores ranging from 32 to 38 indicate a medium risk, and scores ≥ 39 indicate a low risk.

2. The *Flexible Regulation of Emotional Expression* (FREE) scale is a 16-item scenario-based questionnaire assessing flexibility in self-regulatory behaviours [27]. The FREE scale evaluates individuals' perceived ability to modulate their emotional expressions. This component of regulatory flexibility is an important aspect for adjusting to stressful life events. The FREE scale is self-report measure, and the items are rated on a 6-point Likert scale ranging from 1 (unable) to 6 (very able). The FREE scale has three subscales: one measures the ability to enhance emotional expression (FREE Enhance), one measures the ability to suppress emotional expression (FREE Suppress), and one measures overall expressive flexibility (FREE Flexibility). The degree of reliability of the three subscales in the present sample was good, with a Cronbach's α of .77 for the FREE Enhance subscale, .78 for the FREE Suppress subscale, and .79 for the FREE Flexibility subscale.

3. The *Context Sensitivity Index* (CSI) is a 20-item scenario-based questionnaire assessing context sensitivity, which is the ability to perceive cues to contextual demands across different situations [41]. Context sensitivity has been identified as a crucial component of successful self-regulation. The CSI consists of two subscales measuring the ability to capture sensitivity to the presence of contextual cues (CSI Cue Presence index) and sensitivity to the relative absence of cues (Cue Absence index). An overall CSI score is calculated by averaging the Cue Presence and Cue Absence indices. The CSI is a self-report measure, and the items are rated on a 7-point Likert scale ranging from 1 (not at all) to 7 (very much). The degree of reliability of the factors in the present sample was good, with a Cronbach's α of .74 for the Cue Presence index and .72 for the Cue Absence index.

## Statistical analyses

The data were analysed using SPSS v. 22 (IBM, Armonk, NY, USA) statistical software and Excel software v.16.0 (Microsoft Corp. 2016). Data obtained from this study were checked, and descriptive and inferential statistical analyses were then conducted. An independent samples *t*-test was employed to verify any statistically significant gender differences for the observed variables. The Bonferroni correction was applied to address type 1 errors. Gender differences were examined because burnout was often linked to female gender [42, 43], even if these categorizations seem to be unfounded with respect to work-related burnout [44]. The one-sample *t*-test was used to compare the results of this sample with the cut-off score established by the Italian version of the MBI [40].

We conducted a univariate analysis of variance (ANOVA) using the three MBI factors as dependent variables and profession as the independent variable. An identical analysis of covariance (ANCOVA) was subsequently performed, controlling for participant age and work experience. Finally, the FREE and CSI scores were added to the analyses. The CSI subscales, namely, the Cue Presence index and the Cue Absence index, and the FREE subscales, namely, the Enhance and Suppress subscales, were used to verify the Cronbach's alpha scores and gender differences. However, for the purpose of this study, only the FREE Flexibility and CSI overall scores were used for the ANOVA and ANCOVA. Despite the lack of consensus regarding the minimum sample size for conducting multivariate analysis, Bujang and colleagues [45] argued that the ideal sample size is at least 300 for non-experimental studies. However, in this study, ANCOVA was conducted for descriptive model building without the implication of causality between variables [46].

## Results

The demographic characteristics of the sample are shown in Table 1. The study was conducted with a sample of 65 Italian practitioners specializing in palliative care who ranged in age from 22 to 63 years ($M = 36.49 \pm 9.94$). All the participants worked in teams of two, and the palliative home care services were provided predominantly to end-of-life cancer patients.

The final sample consisted of 34 females and 31 males working in two palliative home care services in Sicily, Italy. The average level of formal education was 19.65 years ($SD = 4.06$). Thirty-eight percent of the respondents were specialist palliative care nurses ($n = 25$), 25% were healthcare assistants ($n = 16$), 14% were physiotherapists ($n = 9$), 14% were clinical psychologists ($n = 9$), 6% were social workers ($n = 4$), and 3% were specialists in palliative medicine ($n = 2$). The average work experience s was 17.57 months ($SD = 21.42$). As shown in Table 1, the results from t-tests showed no significant differences between males and females for any of the considered variables.

Statistical comparisons through a series of t-tests with Bonferroni correction revealed no significant gender differences. For twelve comparisons, the new critical alpha level was .004 (.05 divided by 12). These results also showed that the sample was well matched for age,

**Table 1. Demographic characteristics; work experience; and MBI, FREE, and CSI scores of the participants by gender.**

| Variable | Entire sample ($n = 65$) | Male ($n = 31$) | Female ($n = 34$) | t-test |
|---|---|---|---|---|
| Age in years ($SD$) | 36.49 (9.94) | 35.13 (10.39) | 37.74 (9.49) | -1.057 |
| Education level in years ($SD$) | 19.65 (4.06) | 20.03 (3.92) | 19.29 (4.21) | 0.729 |
| Work experience in months ($SD$) | 17.57 (21.42) | 22.29 (28.34) | 13.26 (10.94) | 1.722 |
| MBI EE ($SD$) | 11.65 (9.44) | 12.23 (10.06) | 11.12 (8.96) | 0.470 |
| MBI D ($SD$) | 2.95 (4.00) | 3.03 (4.00) | 2.88 (4.06) | 0.150 |
| MBI PA ($SD$) | 40.34 (6.42) | 39.35 (6.74) | 41.24 (6.08) | -1.183 |
| FREE Enhance ($SD$) | 4.14 (0.73) | 4.21 (0.74) | 4.07 (0.72) | 0.777 |
| FREE Suppress ($SD$) | 3.89 (0.79) | 4.02 (0.85) | 3.77 (0.72) | 1.269 |
| FREE Flexibility ($SD$) | 7.31 (1.50) | 7.36 (1.61) | 7.26 (1.41) | 0.262 |
| CSI Cue Presence ($SD$) | 47.09 (7.29) | 48.61 (7.97) | 45.71 (6.41) | 1.626 |
| CSI Cue Absence ($SD$) | 42.34 (8.57) | 41.19 (9.23) | 43.38 (7.92) | -1.029 |
| CSI ($SD$) | 44.72 (3.72) | 44.90 (3.67) | 44.54 (3.80) | 0.387 |

*$p < 0.05$

**$p < 0.01$

EE = Emotional Exhaustion; D = Depersonalization; PA = Personal Achievement

education level, and work experience, as well as for level of burnout, flexibility, and context sensitivity.

Table 2 illustrates the level of burnout symptoms reported in the entire sample of palliative home care workers and their burnout symptoms based on the Italian version of the MBI. One-sample $t$-tests showed significant differences between scores for the sample of palliative care workers and the cut-off score for burnout symptoms on the Italian MBI [40]. These results were confirmed by descriptive statistics that highlighted a low level of burnout symptoms for the three MBI factors.

Table 3 presents the results of the ANOVA for the three MBI factors. The first ANOVA was carried out using the score of the Emotional Exhaustion subscale of the MBI as the dependent variable and profession as the independent variable. Here, we observed a significant univariate main effect for profession [$F_{(5, 59)} = 2.873$, $p = 0.022$, $n_p^2 = 0.196$], with physiotherapists scoring significantly higher than the other specialist palliative care practitioners on the Emotional Exhaustion subscale. Subsequently, an identical ANCOVA was performed, controlling for participant age and work experience. The results showed a significant univariate effect of participant age [$F_{(1, 57)} = 4.878$, $p = 0.031$, $n_p^2 = 0.079$]. Finally, an identical ANCOVA was conducted with the FREE and CSI scores used as covariates. The results showed that the univariate effects for profession and participant age persisted $F_{(5, 55)} = 6.259$, $p = 0.000$, $n_p^2 = 0.363$ and $F_{(1, 55)} = 6.815$, $p = 0.012$, $n_p^2 = 0.110$, respectively). In addition, a significant univariate effect was found for the FREE scores [$F_{(1, 55)} = 11.084$, $p = 0.002$, $n_p^2 = 0.168$]. The effect of CSI scores on emotional exhaustion approached statistical significance ($p = 0.054$).

An additional ANOVA was conducted using the Depersonalization score as the dependent variable, and no significant univariate effect for profession was found. Moreover, participant age and work experience were not significant covariates. Furthermore, no significant effect was found after adding the FREE and CSI scores as covariates.

A third ANOVA was conducted using the Personal Achievement score as the dependent variable with profession as the independent variable. An identical ANCOVA was run, controlling for participant age and work experience. The FREE and CSI scores were added as covariates as well, but no univariate effect was found.

## Discussion

In the present study, we examined burnout symptoms in a sample of palliative home care practitioners. Sensitivity to context and emotional flexibility were considered to explain burnout symptoms. Personality characteristics (i.e., neuroticism, hardiness, locus of control) might be a relevant factor associated with the development of work distress and burnout [42, 47–52]. In this context, the field of palliative care could be very stressful for all practitioners who are involved in the daily care of terminal patients [53]. In this regard, one study reported a 62 percent burnout rate among palliative care practitioners [4], even though evidence on burnout

**Table 2. Results of burnout symptoms in the sample of palliative home care practitioners.**

| Variable | $M$ | $SD$ | $t$ ($df = 64$) | Burnout symptoms |
|---|---|---|---|---|
| MBI EE | 11.65 | 9.44 | -9.699** | Low |
| MBI D | 2.95 | 4.00 | -6.143** | Low |
| MBI PA | 40.34 | 6.42 | 11.726** | Low |

*$p < 0.05$

**$p < 0.01$

EE = Emotional Exhaustion; D = Depersonalization; PA = Personal Achievement

**Table 3. Results of the ANOVA and ANCOVA for the three factors of the MBI.**

| 1. Emotional Exhaustion | | |
|---|---|---|
| | F | $n_p^2$ |
| Profession | 6.259** | 0.363 |
| Age | 6.815* | 0.110 |
| Work experience | 1.802 | 0.032 |
| FREE | 11.084** | 0.168 |
| CSI | 3.892 | 0.066 |
| 2. Depersonalization | | |
| | F | $n_p^2$ |
| Profession | 1.328 | 0.108 |
| Age | 3.786 | 0.064 |
| Work experience | 0.485 | 0.009 |
| FREE | 0.065 | 0.001 |
| CSI | 1.005 | 0.018 |
| 3. Personal Achievement | | |
| | F | $n_p^2$ |
| Profession | 2.022 | 0.155 |
| Age | 3.044 | 0.052 |
| Work experience | 0.396 | 0.007 |
| FREE | 0.001 | 0.000 |
| CSI | 0.007 | 0.000 |

*$p < 0.05$

**$p < 0.01$

FREE = Flexible Regulation of Emotional Expression; CSI = Context Sensitivity Index

rates is inconsistent [54]. Patients in palliative care units wish for a personal and warm patient-physician relationship, so they expect their symptoms to be attended to with emotional care [55]. On the other hand, the experience of accompanying patients in death can expose palliative care practitioners to burnout risk [11].

In light of this perspective, the first aim of this study was to compare burnout in male and female practitioners specializing in palliative care. Past studies have shown inconsistent results [43, 56], even though burnout was often associated with female gender [41, 42]. In line with the initial hypothesis, the results clearly confirm that females do not have a higher burnout risk than males.

In this context, the second aim of this study was to investigate the risk of burnout among practitioners specializing in palliative home care. In fact, the clinical setting was found to be a significant predictor of burnout, so working on a palliative home care team represents a significant risk factor [8, 9]. Hence, we hypothesized a moderate risk for all symptoms of burnout. In contrast, we found a low burnout risk for emotional exhaustion, depersonalization, and personal achievement. These results could have depended in part on the characteristics of our sample, as the group of practitioners had a mean work experience of approximately 18 months. The relationship between years of experience and burnout has been investigated in depth in numerous research studies [57]. However, evidence from a meta-analysis highlighted only a small negative correlation between work experience and emotional exhaustion [57]. Recently, another study found that years in practice did not distinguish significantly between burned-out and not burned-out specialist palliative care practitioners [58].

In addition, the two palliative home care services in our study had a structured programme in place to support all practitioners. In fact, empirical research has indicated that both individual-focused and structural interventions can reduce burnout among physicians [12]. Unfortunately, interventions to address or prevent burnout in palliative care are still sporadic and mainly focused on individual strategies [59]. Nevertheless, communication among health care providers is often inadequate [60]. From this perspective, coordination staff periodically sought interventions to manage stress and prevent burnout. Structural and organizational interventions adopted to prevent burnout are mainly focused on periodic group discussion, the limitation of weekly working hours for all staff, and the learning of communication skills with patients and colleagues [12]. However, the principal focus of this study is on individual-level factors, and consequently, future studies should also consider organizational factors to explain burnout risk. Despite the notable amount of research conducted in the context of burnout prevention among healthcare practitioners, there is still a lack of studies taking into account both organizational- and individual-level factors [12]. Therefore, further research is needed to evaluate whether these strategies are effective in reducing burnout among specialist palliative care practitioners.

The third aim of this study was to explore the differential contribution of the type of professional practitioner. A majority of studies to date have examined only physicians; however, they have examined only one of the types of professionals found on a palliative care team [54]. For this reason, we decided to examine in more depth the role of profession type in burnout risk. The results showed a significant effect for profession type, with physiotherapists scoring significantly higher than others on emotional exhaustion. One critical factor that may at least partially account for this finding is physiotherapists' level of competence in the communication skills. The physiotherapist often plays a pivotal role on the palliative care team, but further research is needed to examine the impact of communication effectiveness [61, 62]. This effect of profession type persisted even after controlling for participant age and work experience. However, practitioner age also had a significant effect on emotional exhaustion.

The last aim of this study was to explore the role of sensitivity to context and regulatory flexibility on burnout risk. As Bonanno and Burton stated, regulatory flexibility is associated with mental health status and represents an important ability in one's effort to adapt to stressful life events [32]. We hypothesized that the two examined components would have a significant effect on all symptoms of burnout. The results from the ANCOVA indeed revealed a significant effect of regulatory flexibility on emotional exhaustion.

No significant effects were found on the other two burnout factors, depersonalization or personal achievement. These results seem to be consistent with studies that have highlighted different risk factors for the three components of burnout [7]. Moreover, emotional exhaustion showed a strong relationship with individual variables that future research in palliative care should address.

Burnout is a complex syndrome involving several dimensions encompassing both individual and organizational variables. This study's findings can be very useful in the development of more tailored and efficacious interventions to prevent and reduce burnout.

This study has some limitations that should be addressed by future research. First, this study adopted a cross-sectional design that did not allow us to infer causal relationships between the observed variables. Longitudinal studies would better explain the impact of emotional flexibility and sensitivity to context on burnout development among palliative care practitioners. Second, our sample size was small despite the enrolment of 70 percent of all the palliative care practitioners of the two involved services. Therefore, the results may not be generalizable to other palliative home care services.

Third, a control group could be necessary for comparing the obtained results. The fourth limitation regards the use of self-assessment measures to evaluate the risk of burnout, even though the MBI is a well-established and standardized tool in the literature [38–41, 55]. Several previous studies on burnout risk have reported that social desirability could result in the underrating of symptoms [3, 49, 63]. However, the FREE scale and the CSI, which were employed to evaluate regulatory flexibility and context sensitivity, respectively, are scenario-based indices that do not require participants to possess an exact awareness of their own abilities [41]. In fact, these measures were constructed to identify participants' hypothetical ability rather than their recalled history of engagement in regulatory behaviours.

The present study also has relevant strengths. This study is the first attempt to explore the role of sensitivity to context and emotional flexibility in the development of burnout among practitioners working in the palliative home care setting. Moreover, the composition of our sample was highly representative of the different percentages of professional roles working in the two palliative care services. Hence, future research is needed to more thoroughly examine these important findings.

## Conclusion

The results indicated that profession type and age were associated with a higher prevalence of emotional exhaustion among palliative home care practitioners. The results also highlighted a significant effect of regulatory flexibility on emotional exhaustion. To date, this is the first attempt to explore the interplay between emotional flexibility, as a central tenet of resilience, and burnout in palliative home care. Given the relationship between emotional flexibility and burnout, prevention interventions based on resilience could help decrease burnout among palliative home care practitioners. However, there are a number of limitations that should be addressed by future research. A major limitation was the size of the sample. Consequently, the sample may not be representative of other palliative home care practitioners.

## Supporting information

**S1 Database.**
(XLSX)

## Acknowledgments

We thank our research assistant Marilena Vadalà for her help with this project.

## Author Contributions

**Conceptualization:** Vittorio Lenzo, Maria C. Quattropani.

**Data curation:** Vittorio Lenzo, Maria C. Quattropani.

**Formal analysis:** Vittorio Lenzo.

**Investigation:** Vittorio Lenzo, Valentina Bordino.

**Methodology:** Vittorio Lenzo, George A. Bonanno, Maria C. Quattropani.

**Project administration:** Maria C. Quattropani.

**Software:** Vittorio Lenzo.

**Supervision:** George A. Bonanno, Maria C. Quattropani.

**Writing – original draft:** Vittorio Lenzo.

**Writing – review & editing:** Vittorio Lenzo, George A. Bonanno, Maria C. Quattropani.

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
