## [Decision Letter · Decision Letter 0]

20 Dec 2019

PONE-D-19-33880

Understanding the role of regulatory flexibility and context sensitivity in preventing burnout in a palliative home care team

PLOS ONE

Dear Dr Lenzo,

Thank you for submitting your manuscript to PLOS ONE. After careful consideration, we feel that it has merit but does not fully meet PLOS ONE’s publication criteria as it currently stands. Therefore, we invite you to submit a revised version of the manuscript that addresses the points raised during the review process.

We would appreciate receiving your revised manuscript by Feb 03 2020 11:59PM. To enhance the reproducibility of your results, we recommend that if applicable you deposit your laboratory protocols in protocols.io, where a protocol can be assigned its own identifier (DOI) such that it can be cited independently in the future. For instructions see: http://journals.plos.org/plosone/s/submission-guidelines#loc-laboratory-protocols

We look forward to receiving your revised manuscript.

Kind regards,

Tim Luckett

Academic Editor

PLOS ONE

Reviewers' comments:

Reviewer's Responses to Questions

**Comments to the Author**

1. Is the manuscript technically sound, and do the data support the conclusions?

Reviewer #1: Partly

Reviewer #2: Partly

2. Has the statistical analysis been performed appropriately and rigorously? 

Reviewer #1: No

Reviewer #2: No

3. Have the authors made all data underlying the findings in their manuscript fully available?

Reviewer #1: Yes

Reviewer #2: No

4. Is the manuscript presented in an intelligible fashion and written in standard English?

Reviewer #1: Yes

Reviewer #2: No

5. Review Comments to the Author

Reviewer #1: 1) This is paper should be analyzed and included by the authors: Rizo-Baeza M, Mendiola-Infante SV, Sepehri A, Palazón-Bru A, Gil-Guillén VF, Cortés-Castell E. Burnout syndrome in nurses working in palliative care units: An analysis of associated factors. J Nurs Manag. 2018;26:19-25. doi: 10.1111/jonm.12506.

2) Abstract: you should include this information:

a. Background.

b. The country and dates of the study.

c. A summary of the statistical methods.

d. The factors which you analyzed.

3) Introduction should be shortened a lot. In general a scientific paper has a maximum of 800 words for this section.

4) Which hospitals were included? You only indicated Sicilia.

5) “The consent form informed participants of their right to withdraw from the study.” Patients?

6) More information is necessary for the sample size calculation: method and design effect and their parameters.

7) Profession type was not defined, as well as the other covariables not based on questionnaires.

8) Please, take into account that you can only include six predictors in the multivariate models.

9) I will review the rest of the paper once the authors apply my own comments.

Reviewer #2: Please justify the use of MANCOVA with such a small sample. See Bujang, M. A., Sa’at, N., & Bakar, T. M. I. T. A. (2017). Determination of minimum sample size requirement for multiple linear regression and analysis of covariance based on experimental and non-experimental studies. Epidemiology, Biostatistics and Public Health, 14(3).

Results

Table 1 Why were you examining differences in study measures by gender?

Table 2 Why did you conduct multiple t-tests? How did you address risk of type 1 errors?

Discussion

Personality traits can be considered as vulnerability factors in the

development of work distress and burnout. Provide references to support this statement.

Field and environment are two different things. Do you mean that the practice of the profession is stressful? If so, what aspects?

Operational tasks linked to diagnostic and therapeutic practices,

however, should be considered as strictly related to psychological tasks just as maintaining concordance among patients and their relatives [43]. This sentence is ambiguous. Do you mean there is a psychological component to practice for the health care professional?

The relationship between years of experience and burnout has been

investigated in-depth across numerous research studies - references to support this statement.

Structural and organisational interventions adopted to prevent burnout are mainly focused on periodical group discussion, limit of weekly working hours for all staff, and learning communication skills with patients and colleagues. - references to support statement.

Why are you reporting on structural factors in your discussion when you are looking at individual level factors?

Please provide reference for the limitations of self-assessment measures.

Please provide a conclusion section to this paper.

6. PLOS authors have the option to publish the peer review history of their article (what does this mean?). If published, this will include your full peer review and any attached files.

Reviewer #1: No

Reviewer #2: No

---

## [Author Response · Author response to Decision Letter 0]

3 Mar 2020

Authors' Responses to Reviewer's Comments

Reviewer #1: 

Reply: We thank the reviewer for careful reading of the manuscript. We have wisely taken the comments on board to ameliorate and make clear the manuscript. Please, see below a detailed point-by-point response to all comments (reviewer’s comments in violet, our replies in red). 

1) This is paper should be analyzed and included by the authors: Rizo-Baeza M, Mendiola-Infante SV, Sepehri A, Palazón-Bru A, Gil-Guillén VF, Cortés-Castell E. Burnout syndrome in nurses working in palliative care units: An analysis of associated factors. J Nurs Manag. 2018;26:19-25. doi: 10.1111/jonm.12506.

Reply: We want to thank the reviewer for the suggested article that is very interesting for our study. So, we have included citation to the Rizo-Baeza and colleagues’ paper to acknowledge this original work in addition to the others papers. In this regard, we have added three sentences to highlight the following critical point in the literature: (a) the lack of research evaluating burnout in nurses; (b) the need for research to provide goodness of fit of statistical models.

Consequently, we have attentionally revised studies on burnout among palliative care practitioners. 

However, we feel that some points of weakness are strictly related to complexity of research in palliative care. For example, it is often difficult to find well-balanced sample demographic and work-related variables (e.g., gender).

2) Abstract: you should include this information:

a. Background.

b. The country and dates of the study.

c. A summary of the statistical methods.

d. The factors which you analyzed.

Reply: We have included this information in the abstract, as kindly suggested.

3) Introduction should be shortened a lot. In general a scientific paper has a maximum of 800 words for this section.

Reply: We agree that introduction should be shortened a lot. We have now restructured the introduction section in order to be about 800 words. Now, introduction section comprises 804 words.

4) Which hospitals were included? You only indicated Sicilia.

Reply: We apologise for this vague information. So, we have provided additional information on structures where this research took place. Two palliative care reference services operating in the regional territory belonging to the Italian National Health Service (INHS) were involved in this study. We would point out that the two services deliver exclusively palliative home care. These palliative home care services assist about eight hundred patients and their caregivers every year.

5) “The consent form informed participants of their right to withdraw from the study.” Patients?

Reply: Sorry. We have removed the sentence from the manuscript.

6) More information is necessary for the sample size calculation: method and design effect and their parameters.

Reply: Many thanks for the opportunity to improve and clarify this point. We have carefully considered this issue on the sample size calculation and statistical power. Hence, following the Cohen’s guidelines (1988), we decided to reconsider our power analysis. In fact, application of analysis of covariance in our research was for descriptive model building, without implication of causality because it is not substitute for conducting experiment (Tabachnick and Fidell, 2013).

Consequently, we calculated statistical power with post hoc type analysis (Cohen, 1988). Therefore, statistical power is computed as a function of significance level α, sample size, and population effect size. More specifically, we obtained a statistical power of .88 (with a critical F of 4.01) by inserting a significant finding (at the .05 level), a large effect size (Cohen’s d = 0.40), a total sample size of 65, 6 number of groups, and 4 covariates. Undoubtedly, we selected F test and ANCOVA with fixed effects, main effects and interactions, as settings.

7) Profession type was not defined, as well as the other covariables not based on questionnaires.

Reply: We feel the reviewer may have misconceived us here. Profession type, age, and work experience were defined in results section. Since this premise a more deeply delineation of the sample was done (i.e., number and not only percentage for profession type was added). In addition, we have acknowledged in the discussion section that we focalized on individual level factors and hence this limitation should be addressed by future research. 

8) Please, take into account that you can only include six predictors in the multivariate models.

Reply: Certainly, we have conducted three ANOVA using the MBI factors as dependent variables with profession as the only independent variable. Subsequently, an identical ANCOVA was carried out with two covariates (age and work experience). Lastly, we performed ANCOVA by adding FREE and CSI. So, the final model included 1 independent variable and four covariates.

9) I will review the rest of the paper once the authors apply my own comments.

Reply: Thank you so much for the time invested and the important contributions suggested, which have helped us to improve the quality of the manuscript. We hope that you give us next suggestions to improve the manuscript.

Reviewer #2: 

Reply: We thank the reviewer for constructive remarks and important considerations about the manuscript. We have sagely taken the comments on board improve the manuscript. Please find below a detailed point-by-point response to all comments (reviewers’ comments in black, our replies in blue).

Please justify the use of MANCOVA with such a small sample. See Bujang, M. A., Sa’at, N., & Bakar, T. M. I. T. A. (2017). Determination of minimum sample size requirement for multiple linear regression and analysis of covariance based on experimental and non-experimental studies. Epidemiology, Biostatistics and Public Health, 14(3).

Reply: Thank you for your precious suggestion related to the use of analysis of covariance with a small sample. We find very interesting the paper by Bujang and colleagues (2017) for the simplicity and useful and we have inserted it in the manuscript. We will also take it into consideration for our next research.

The authors stated that the ideal sample size shall preferably be at least 300 for non-experimental studies. At this regard, we would to highlight that application of analysis of covariance in our research was for descriptive model building. In light of this perspective, for instance, Tabachnick and Fidell (2013) pointed out that the covariances improve prediction of the dependent variable, without implication of causality because analysis of covariance is not a substitute for conducting an experiment. In addition, burnout research in palliative care is strictly related to some methodological inherent difficulties, especially in home care setting (e.g. in our research two samples were involved but the sample size was 67). At this regard, several studies have utilized small sample, even after power analysis. Moreover, most of them have conducted regression analyses or others multivariate techniques. Please see, for example, the following papers:

- Coffey, M. (1999). Stress and burnout in forensic community mental health nurses: an investigation of its causes and effects. Journal of Psychiatric and Mental Health Nursing, 6(6), 433–443. doi:10.1046/j.1365-2850.1999.00243.x

- Galeazzi, G. M., Delmonte, S., Fakhoury, W., & Priebe, S. (2004). Morale of mental health professionals in Community Mental Health Services of a Northern Italian Province. Epidemiologia e Psichiatria Sociale, 13(3), 191–197. doi:10.1017/s1121189x00003456

- Dunwoodie, D. A., & Auret, K. (2007). Psychological morbidity and burnout in palliative care doctors in Western Australia. Internal Medicine Journal, 0(0), 070521010100003–??? doi:10.1111/j.1445-5994.2007.01384.x

- Bressi, C., Porcellana, M., Gambini, O., Madia, L., Muffatti, R., Peirone, A., … Altamura, A. C. (2009). Burnout Among Psychiatrists in Milan: A Multicenter Survey. Psychiatric Services, 60(7), 985–988. doi:10.1176/ps.2009.60.7.985

- Madathil, R., Heck, N. C., & Schuldberg, D. (2014). Burnout in Psychiatric Nursing: Examining the Interplay of Autonomy, Leadership Style, and Depressive Symptoms. Archives of Psychiatric Nursing, 28(3), 160–166. doi:10.1016/j.apnu.2014.01.002

Consequently, on the basis of these considerations, we have clarified the use of analysis of covariance in the statistical analyses section. Moreover, we have deeply recognized in the discussion section that the results may not generalize. Finally, we have underlined in the discussion section that the cross-sectional design of this study did not allow to make inference of causality between the observed variables. 

Results

Table 1 Why were you examining differences in study measures by gender?

Reply: Many thanks for the opportunity to explain this point. Several past studies have examined the relationship between burnout and gender (for a recent review see the following papers:

- Purvanova RK, Muros JP. Gender differences in burnout: A meta-analysis. J Vocat Behav. 2010;77(2):168–85. do: http://dx.doi.org/10.1016/j.jvb.2010.04.006

- O’Connor K, Muller Neff D, Pitman S. Burnout in mental health professionals: A systematic review and meta-analysis of prevalence and determinants. Eur Psychiatry. 2018; 53:74–99. doi:10.1016/j.eurpsy.2018.06.003) with inconsistent results. 

In addition, burnout was frequently associated to female gender (see for example the following papers: 

 - Maslach C, Schaufeli WB, Leiter MP. Job Burnout. Annu Rev Psychol. 2001;52(1):397–422. doi: http://dx.doi.org/10.1146/annurev.psych.52.1.397

- Burns KEA, Fox-Robichaud A, Lorens E, Martin CM. Gender differences in career satisfaction, moral distress, and incivility: a national, cross-sectional survey of Canadian critical care physicians. Can J Anaesth. 2019;66(5):503–11. doi: http://dx.doi.org/10.1007/s12630-019-01321-y).

For these reasons, we decided to examine gender differences for the burnout factors. Results from this well-balanced sample for gender (31 males vs 35 females without significant differences for age, education, and work experience) have not shown differences for any subscales of the MBI.

We have briefly discussed these results in the discussion section with the aim to make more easily understandable this choice for readers. Consequently, we have added this comparison as the first aim of this study. We feel that this change did not influence the nature of our work because comparison by gender was already inserted in the statistical section and results sections.

Table 2 Why did you conduct multiple t-tests? How did you address risk of type 1 errors?

Reply: As stated above, multiple t-tests were conducted to examine gender difference. We fully understand the reviewer’s viewpoint about risk of type 1 errors. Therefore, we have adopted a conservative approach using the Bonferroni correction. 

Discussion

Personality traits can be considered as vulnerability factors in the

development of work distress and burnout. Provide references to support this statement.

Reply: We revised the terminology, in order to avoid misinterpretation, and added references. Actually, cited studies were mostly designed as cross-sectional. In this vein, we think it could be speculative to talk about vulnerability factors. This has been emphasized in the revised manuscript.

Hence, we have replaced the phrase above with the follow and provided references to support:

“Personality characteristics (i.e., neuroticism, hardiness, locus of control) might be a relevant factor associated with the development of work distress and burnout [42,47-52].

42. Maslach C, Schaufeli WB, Leiter MP. Job Burnout. Annu Rev Psychol. 2001;52(1):397–422. doi: http://dx.doi.org/10.1146/annurev.psych.52.1.397

47. Purvanova RK, Muros JP. Gender differences in burnout: A meta-analysis. J Vocat Behav. 2010;77(2):168–85. do: http://dx.doi.org/10.1016/j.jvb.2010.04.006

48. Swider BW, Zimmerman RD. Born to burnout: A meta-analytic path model of personality, job burnout, and work outcomes. J Vocat Behav. 2010; 76(3):487–506. doi:10.1016/j.jvb.2010.01.003

49. Armon G, Shirom A, Melamed S. The Big Five Personality Factors as Predictors of Changes Across Time in Burnout and Its Facets. J Pers. 2012; 80(2): 403–27. doi:10.1111/j.1467-6494.2011.00731.x

50. Armon G. Type D personality and job burnout: The moderating role of physical activity. Pers Individ Differ. 2014; 58:112–5. doi:10.1016/j.paid.2013.10.020

51. De la Fuente-Solana EI, Gómez-Urquiza JL, Cañadas GR, Albendín-García L, Ortega-Campos E, Cañadas-De la Fuente GA. Burnout and its relationship with personality factors in oncology nurses. Eur J Oncol Nurs. 2017; 30: 91–6. doi:10.1016/j.ejon.2017.08.004

52. Gama G, Barbosa F, Vieira M. Personal determinants of nurses’ burnout in end of life care. Eur J Oncol Nurs. 2014; 18(5):527-33. doi: 10.1016/j.ejon.2014.04.005

Field and environment are two different things. Do you mean that the practice of the profession is stressful? If so, what aspects?

Reply: Certainly, we have amended this part to be clearer utilizing the word “field” across the manuscript. We mean just that professional practice is stressful. In fact, burnout is frequent in fields of medicine with high patient mortality rates. For instance, Kamal and colleagues [4] reported a burnout rate of 62 percent among hospice and palliative care clinicians. However, this is one of the highest rates reported among palliative care. 

Operational tasks linked to diagnostic and therapeutic practices,

however, should be considered as strictly related to psychological tasks just as maintaining concordance among patients and their relatives [43]. This sentence is ambiguous. Do you mean there is a psychological component to practice for the health care professional?

Reply: We agree with the reviewer that this sentence is unclear. Undoubtedly, psychological component plays a fundamental role in palliative home care practice and impacts patients’ quality of life. So, we have revised this part to point out the significance of the psychological aspects in palliative home care. On the other hand, we have marked psychological effect for palliative care practitioners. For these reasons, we have included citations to the Masel and colleagues (2016) and to the Rugnone and colleagues (2017) papers. 

The relationship between years of experience and burnout has been

investigated in-depth across numerous research studies - references to support this statement.

Reply: Many thanks to the reviewer for this precious indication. As stated, this relationship has been examined in several papers. Therefore, we include citation to a recent systematic review including 33 studies (O’Connor K, Muller Neff D, Pitman S. Burnout in mental health professionals: A systematic review and meta-analysis of prevalence and determinants. Eur Psychiatry. 2018; 53:74–99. doi:10.1016/j.eurpsy.2018.06.003). It is to be emphasized that the relationship between years of experience and burnout was not consistent across the studies. Consequently, we have included citations to the Brewer and colleagues (2004) and Marchalik and colleagues (2019) papers to point out this question.

Structural and organisational interventions adopted to prevent burnout are mainly focused on periodical group discussion, limit of weekly working hours for all staff, and learning communication skills with patients and colleagues. - references to support statement.

Reply: We have added the following reference to support statement:

12. West CP, Dyrbye LN, Erwin PJ, Shanafelt TD. Interventions to prevent and reduce physician burnout: a systematic review and meta-analysis. The Lancet. 2016;388(10057):2272–81. doi: http://dx.doi.org/10.1016/s0140-6736(16)31279-x

Why are you reporting on structural factors in your discussion when you are looking at individual level factors?

Reply: We agree that it is essential to make this distinction clear. So, we apologize for the confusion. We have amended this part of discussion by clarifying we focalized only on individual level factors. In fact, this is now better explained as follow: 

“Structural and organisational interventions adopted to prevent burnout are mainly focused on periodical group discussion, limit of weekly working hours for all staff, and learning communication skills with patients and colleagues [12]. However, the principal focus of this study is on individual level factors and consequently future studies should also consider organisational factors to explain burnout risk. Despite the notable amount of research conducted in the context of burnout prevention among healthcare practitioners, there is still a lack of studies taking account both organisational and individual level factors [12]. Therefore, further research is needed to evaluated if these strategies are effective to reduce burnout among specialist palliative care practitioners.”

Please provide reference for the limitations of self-assessment measures.

Reply: We have provided additional information to clarify limitations of self-assessment measures in this study. In fact, self-report measures can be influenced by social desirability, and for this reason, past studies have indicated that social desirability can result in underreporting of symptoms of burnout. This is highly probable for smaller organizations such as those involved in our study. Hence, we have restructured the part of manuscript as follow: “The fourth limitation regards the use of self-assessment measures to evaluate the risk of burnout even though the MBI is a well-established and standardised tool in the literature [36-39,51]. Several previous studies on burnout risk have reported that social desirability could result in underrating of symptoms [3,47,59]. However, the FREE and the CSI, employed to evaluate regulatory flexibility and context sensitivity, are scenario-based indices that did not require participants to possess an exact awareness of their own abilities [39]. In fact, these measures were constructed to identify their hypothetical ability rather than their reminisced history of engaging in regulatory behaviours.” 

3. Koh MYH, Chong PH, Neo PSH, Ong YJ, Yong WC, Ong WY, et al. Burnout, psychological morbidity and use of coping mechanisms among palliative care practitioners: A multi-centre cross-sectional study. Palliat Med. 2015;29(7):633–42. doi: http://dx.doi.org/10.1177/0269216315575850

39. Bonanno GA, Maccallum F, Malgaroli M, Hou WK. The Context Sensitivity Index (CSI): Measuring the Ability to Identify the Presence and Absence of Stressor Context Cues. Assessment. 2018;107319111882013. doi: 10.1177/1073191118820131

47. Armon G. Type D personality and job burnout: The moderating role of physical activity. Pers Individ Differ. 2014; 58:112–5. doi:10.1016/j.paid.2013.10.020

49. Lapa TA, Carvalho SA, Pinto-Gouveia J. Psychological distress, burnout and personality traits in Dutch anaesthesiologists. Eur J Anaesthesiol. 2017; 34(1) 41–42. doi:10.1097/eja.0000000000000500 

Please provide a conclusion section to this paper.

Reply: A conclusion section restating the main findings and limitations was added.

Language was deeply revised by authors, as preciously suggested by the Reviewer.

Nevertheless, authors kindly point out that they previously received assistance from a professional editing service.

Thank you so much for the time invested and the important contributions suggested, which have helped us to ameliorate the quality of the manuscript.

---

## [Decision Letter · Decision Letter 1]

13 Mar 2020

PONE-D-19-33880R1

Understanding the role of regulatory flexibility and context sensitivity in preventing burnout in a palliative home care team

PLOS ONE

Dear Dr Lenzo,

Thank you for submitting your manuscript to PLOS ONE. I am pleased to confirm that you have successfully addressed the reviewers' comment. However, your submission still requires substantial editing for English grammar and usage. We ask that you please have the manuscript copy-edited by either a native-English speaking colleague or a professional copy-editing service. While you may approach any qualified individual or any professional scientific editing service of your choice, PLOS has partnered with American Journal Experts (AJE) to provide discounted services to PLOS authors. AJE has extensive experience helping authors meet PLOS guidelines and can provide language editing, translation, manuscript formatting, and figure formatting to ensure your manuscript meets our submission guidelines. If the PLOS editorial team finds any language issues in text that AJE has edited, AJE will re-edit the text for free. To take advantage of this special partnership, use the following link: https://www.aje.com/go/plos/."<o:p></o:p>

We would appreciate receiving your revised manuscript by Apr 27 2020 11:59PM. To enhance the reproducibility of your results, we recommend that if applicable you deposit your laboratory protocols in protocols.io, where a protocol can be assigned its own identifier (DOI) such that it can be cited independently in the future. For instructions see: http://journals.plos.org/plosone/s/submission-guidelines#loc-laboratory-protocols

We look forward to receiving your revised manuscript.

Kind regards,

Tim Luckett

Academic Editor

PLOS ONE

---

## [Author Response · Author response to Decision Letter 1]

27 Apr 2020

Vittorio Lenzo

University of Cassino and South Latium

Località Folcara

Cassino, Lazio, Italy 03043

vlenzo@unime.it

Tim Luckett

Academic Editor

PLOS ONE

April 27, 2020

Dear Dr. Luckett:

I am pleased to submit the revised version of the research article entitled “Understanding the role of regulatory flexibility and context sensitivity in preventing burnout in a palliative home care team (PONE-D-19-33880R1)” for publication in PLOS ONE.

As you requested, we sent the manuscript for substantial editing for English grammar and usage. We have included the Editing Certificate by the America Journal Experts (AJE). Also, we carefully revised the manuscript for any error once again. We added the full citation of the using statistical program and of an article we have forgotten in the last revision.

There is no change in our conflicts of interest disclosure.

Once again, we want to thank you for your valuable support and precious support.

Sincerely,

Vittorio Lenzo, PhD

Post-doctoral research fellow, Department of Human, Social and Health Sciences

University of Cassino and South Latium

---

## [Editor Report · Decision Letter 2]

30 Apr 2020

Understanding the role of regulatory flexibility and context sensitivity in preventing burnout in a palliative home care team

PONE-D-19-33880R2

Dear Dr. Lenzo,

We are pleased to inform you that your manuscript has been judged scientifically suitable for publication and will be formally accepted for publication once it complies with all outstanding technical requirements.

With kind regards,

Tim Luckett

Academic Editor

PLOS ONE

---

## [Editor Report · Acceptance letter]

7 May 2020

PONE-D-19-33880R2 

Understanding the role of regulatory flexibility and context sensitivity in preventing burnout in a palliative home care team 

Dear Dr. Lenzo:

I am pleased to inform you that your manuscript has been deemed suitable for publication in PLOS ONE. Congratulations! Your manuscript is now with our production department. 

With kind regards,

on behalf of

Dr. Tim Luckett 

Academic Editor

PLOS ONE